# Genetic Variant c.245A>G (p.Asn82Ser) in *GIPC3* Gene Is a Frequent Cause of Hereditary Nonsyndromic Sensorineural Hearing Loss in Chuvash Population

**DOI:** 10.3390/genes12060820

**Published:** 2021-05-27

**Authors:** Nika V. Petrova, Andrey V. Marakhonov, Natalia V. Balinova, Anna V. Abrukova, Fedor A. Konovalov, Sergey I. Kutsev, Rena A. Zinchenko

**Affiliations:** 1Research Centre for Medical Genetics, 115522 Moscow, Russia; npetrova63@mail.ru (N.V.P.); balinovs@mail.ru (N.V.B.); kutsev@mail.ru (S.I.K.); renazinchenko@mail.ru (R.A.Z.); 2Presidential Perinatal Center of the Public Health Ministry of Chuvashia, Genetic Counseling Department, 428018 Cheboksary, Russia; metra2009@yandex.ru; 3Independent Clinical Bioinformatics Laboratory, 123181 Moscow, Russia; info@clinbio.ru; 4N.A. Semashko National Research Institute of Public Health, 105064 Moscow, Russia

**Keywords:** hearing loss, deafness, Chuvash population, *GIPC3*, population frequency, “bottle neck” effect

## Abstract

Hereditary nonsyndromic sensorineural hearing loss is a disease in which hearing loss occurs due to damage to the organ of the inner ear, the auditory nerve, or the center in the brain that is responsible for the perception of sound, characterized by wide locus and allelic heterogeneity and different types of inheritance. Given the diversity of population of the Russian Federation, it seems necessary to study the ethnic characteristics of the molecular causes of the disease. The aim is to study the molecular and genetic causes of hereditary sensorineural hearing loss in Chuvash, the fifth largest ethnic group in Russia. DNA samples of 26 patients from 21 unrelated Chuvash families from the Republic of Chuvashia, in whom the diagnosis of hereditary sensorineural hearing loss had been established, were analyzed using a combination of targeted Sanger sequencing, multiplex ligase-dependent probe amplification, and whole exome sequencing. The homozygous variant NM_133261.3(*GIPC3*):c.245A>G (p.Asn82Ser) is the major molecular cause of hereditary sensorineural hearing loss in 23% of Chuvash patients (OMIM #601869). Its frequency was 25% in patients and 1.1% in healthy Chuvash population. Genotyping of the NM_133261.3(*GIPC3*):c.245A>G (p.Asn82Ser) variant in five neighboring populations from the Volga-Ural region (Russian, Udmurt, Mary, Tatar, Bushkir) found no evidence that this variant is common in those populations.

## 1. Introduction

Congenital hearing loss is one of the most frequent human diseases, occurring in 1–2 out of 1000 newborns, among which hereditary forms account for more than 50% [1,2,3]. According to the World Health Organization, about 360 million people, which is about 5% of the world’s population, have hearing loss leading to disability, and 32 million of them are children [4]. Sensorineural hearing loss (91.4% of patients) prevails in children’s prelingual isolated hearing loss; mixed hearing loss is found in 7.1% of patients and conductive hearing loss is found in only 1.5% of cases [5].

Most of the hereditary hearing loss forms (about 70%) are nonsyndromic, characterized by wide locus and allelic heterogeneity. To date, more than 6000 pathogenic variants associated with nonsyndromic hearing loss (NSHL) have been identified in more than 120 genes.

The structure of hereditary nonsyndromic hearing loss is dominated by autosomal recessive (AR) forms (70–80%). The share of autosomal dominant (AD) accounts for up to 10–20%; X-linked (1–2%) and mitochondrial (about 1%) forms of hearing loss are relatively rare [5,6,7].

Pathogenic variants in the same genes may be the cause of both AD and AR hearing loss, and digenic inheritance of the disease has been described [7]. In general, 20–30% of hereditary hearing disorders are detected as part of various hereditary syndromes, i.e., hearing loss is accompanied by damage to other organs and systems which are not always simultaneously manifested in the patient’s clinical picture [8,9,10]. More than 400 hereditary syndromes with hearing loss have been described.

Early detection of hearing impairment and subsequent clinical and accurate DNA diagnostics of the genetic form of patient’s hearing loss allows medical practitioners to determine the cause of the disease at an early age, and the timely adoption of measures for the rehabilitation of children with hearing impairment determines the success of their social adaptation.

Hereditary nonsyndromic hearing loss (NSHL) is a disease in which hearing loss occurs due to damage to the organs of the inner ear, the auditory nerve, or the center in the brain that is responsible for the perception of sound. Mapping of the genes responsible for hearing loss was undoubtedly a breakthrough in understanding molecular mechanisms of hearing loss. Mutations in the *GJB2* gene have shown to be the most common cause of NSHL [11]. On average, about 50% of AR NSHL cases in European populations are associated with *GJB2* mutations [12]. The highest contribution of *GJB2* pathogenic variants was shown in European countries (27.1%), while the lowest contribution was in sub-Saharan Africa (5.6%) [13]. In most European countries, the most common pathogenic *GJB2* variant in AR NSHL patients was c.35delG [13,14]. In the Russian population the c.35delG variant also prevails in NSHL (81% of *GJB2* mutant alleles) [2]. The variants in genes *STRC*, *USH2A*, *SLC26A4*, *MYO7A*, *OTOF*, *MYO15A*, and *TECTA* found both in Europe and worldwide were less frequent. Mutations in other genes were observed, as a rule, in individual cases [10].

Given that the population of the Russian Federation is represented by many ethnic groups, it is necessary to study different population-specific molecular causes. This is currently being carried out by the staff of the Laboratory of Genetic Epidemiology for other common hereditary diseases [15,16,17,18].

Our previous studies have shown that both allelic and locus heterogeneity of NSHL was observed in populations and ethnic groups of the Russian Federation [19,20,21,22,23,24,25,26]. For example, in Karachay population the frequency of c.35delG mutation was 0.14%, in Bashkirs and Udmurts—0.25%, in Chuvash—0.48%, with the same birth prevalence of the disease as in Russians, 1:2000–2500. Sequencing of the *GJB2* gene allowed further identification of only 1–2 mutations with extremely low frequencies.

The study of molecular and genetic causes of NSHL in Chuvash, the fifth largest ethnic group in the Russian Federation, living mainly in the Republic of Chuvashia, inhabiting the east of the East European Plain, was performed.

## 2. Materials and Methods

### 2.1. Ethical Statement

Written informed consent was obtained from participants of the 21 Chuvash families (consent from parents was obtained in the case of minors under the age of 18). The study was approved by the local Ethics Committee of the Research Centre for Medical Genetics (RCMG), Moscow, Russian Federation (Protocol No.17/2006 dated 2 February 2006 and Protocol No.5/2010 dated 20 December 2010).

### 2.2. Characteristics of Patients

In total, 26 patients with NSHL from 21 unrelated families were examined. All patients and their parents are Chuvash from the Republic of Chuvashia. Ethnicity up to the third generation was established in the survey. All patients were examined by a surdologist and had bilateral (except for family #17) prelingual sensorineural hearing loss of severe to profound or complete deafness. In the patients’ medical history, there were no possible external environmental risk factors for the development of hearing loss. The diagnosis was made on the basis of the clinical presentation in the surdological center of Cheboksary city. The patients were also examined by a geneticist at the RCMG in order to exclude syndromic forms of hearing loss.

The average age of patients at the time of the examination was 18.57±2.10 years (SD 10.72; range 1.00–42.00). In all the examined patients, venous blood was collected for molecular genetic research. The clinical information on probands is summarized in Table 1.

### 2.3. Characteristics of Healthy Individuals

DNA samples of healthy unrelated individuals from six neighboring populations of the Volga-Ural region (Russians from Kirov region, Mari, Udmurts, Chuvash, Tatars, Bashkirs) from the biobank of the Laboratory of Genetic Epidemiology, collected during genetic and epidemiological expeditions in 2000–2015, were used.

### 2.4. Molecular Genetic Methods

#### 2.4.1. Sequencing of *GJB2*, *GJB3*, and *GJB6* Genes

Sanger sequencing of noncoding and coding exons of *GJB2* gene and coding exons of *GJB3* and *GJB6* genes was performed in 26 patients as described previously [2]. The sequences of the primers used are shown in Table 2. Sequence chromatogram analysis was performed using the ChromasPro Version 1.42 software (Technolysium Pty Ltd., Queensland, Australia).

#### 2.4.2. Multiplex Ligase-Dependent Probe Amplification (MLPA) Analysis

The search for copy number variations (CNV) affecting loci of *GJB3* (1p34.3), *WFS1* (4p16.1), *GJB2* (13q12.11), *GJB6* (13q12.11), *POU3F4* (Xq21.1) genes was performed in 26 patients by multiplex ligase-dependent amplification of samples (MLPA) using the SALSA MLPA Probemix P163-D1 (MRC Holland, Amsterdam, The Netherlands) reagent kit, in accordance with the manufacturer’s protocol. The interpretation of MLPA results was carried out using the Coffalyzer v.140721.1958 software (MRC Holland, Amsterdam, The Netherlands).

#### 2.4.3. Whole Exome Sequencing

Whole exome sequencing (WES) was performed on Illumina HiSeq 2000 instrument (Illumina, San Diego, CA, USA) in 2 × 100 bp paired-end mode in the Genotek Ltd. (Moscow, Russia). A bioinformatics pipeline of WES data analysis was performed as described previously [27]. Further filtering was performed by functional consequences, population frequencies as well as clinical relevance. All variants were named according to the NM_133261.3(*GIPC3*_v001) reference transcript variant. NGS findings of definite/probable diagnostic value were verified by Sanger sequencing in patient, sibling, and parents (family #07).

#### 2.4.4. Sequencing of *GIPC3* Gene Coding Region

Sanger sequencing of 6 exons and exon–intron junction regions of the *GIPC3* gene in 22 patients was performed using the primers presented in Table 2.

#### 2.4.5. Analysis for the NM_133261.3(*GIPC3*):c.245A>G (p.Asn82Ser) Variant in *GIPC3* Gene

The population screening for the NM_133261.3(*GIPC3*):c.245A>G (p.Asn82Ser) variant in *GIPC3* gene was carried out by polymerase chain reaction followed by restriction fragment length polymorphism analysis (PCR-RFLP). For PCR, primers, direct GIPC3-Fmut and reverse GIPC3-Rmut (Table 2) were used. The length of the amplification product was 532 bp. After restriction by endonuclease *Bst*DEI (Sibenzyme, Moscow, Russia), in the case of the normal allele, fragments—298 bp and 234 bp—were formed; in the case of the variant NM_133261.3(*GIPC3*):c.245A>G (p.Asn82Ser)—192 bp, 104 bp, and 234 bp.

The frequency of identified allele was calculated according to the formula: p_i_ = n_i_/n, where n_i_ is the number of i-th allele, n is the sample size (the number of tested chromosomes) [28]. The Exact test was used for calculating 95% confidence intervals (95% CI) [29].

## 3. Results

At first, sequencing of coding regions of *GJB2*, *GJB6*, and *GJB3* genes and analysis of copy number variations affecting loci of *GJB3*, *WFS1*, *GJB2*, *GJB6*, *POU3F4* genes were carried out in 26 patients with NSHL. No pathogenic changes were detected in any patient.

Then, DNA of patient III-1 from family #07, in which both parents and sibling were affected by NSHL (congenital deafness of 4th degree of severity) (Figure 1) was analyzed by whole exome sequencing (WES). All grandparents had no hearing impairment, but deafness was observed in both parents and their siblings.

A homozygous single nucleotide variant NM_133261.3(*GIPC3*):c.245A>G (p.Asn82Ser) in *GIPC3* gene leading to a missense substitution (p.Asn82Ser) was identified. This substitution affects the conservative position within the N-terminal GIPC homologous domain 1 (GH1). The frequency of the variant in gnomAD database is 0.00003596 (in 9 affected chromosomes out of 250288); no homozygous carriers were registered [28]. Most pathogenicity prediction tools, such as SIFT (https://sift.bii.a-star.edu.sg/ accessed on 26 February 2021), Polyphen2 (http://genetics.bwh.harvard.edu/pph2/ accessed on 26 February 2021), and MutationTaster (http://www.mutationtaster.org/ accessed on 26 February 2021), classify the variant as damaging. The genetic variant identified by WES was verified by Sanger sequencing in family #07. Sanger sequencing of the *GIPC3* gene fragment in the affected subjects of family #07 (parents and siblings) revealed that all of them carried c.245A>G variant in the homozygous state.

After that, bidirectional sequencing of all six exons of the *GIPC3* gene was performed in 22 Chuvash patients with NSHL (Figure 2). Additionally, two patients, homozygous for variant c.245A>G (families #01 and #20), and one patient, heterozygous for variant c.245A>G in the *GIPC3* gene (family #04), were identified. The total number of homozygous patients appeared to be six, with one additional patient carrying this variant in the heterozygous state (Table 1). Thus, the frequency of NM_133261.3(*GIPC3*):c.245A>G (p.Asn82Ser) variant in Chuvash NSHL patients was 25.0% (13/52 chromosomes). Based on sequencing results of the entire *GIPC3* gene it became possible to determine the haplotype of common single nucleotide polymorphisms (SNP) linked to the NM_133261.3(*GIPC3*):c.245A>G (p.Asn82Ser) variant in the *GIPC3* gene. In all cases, it was the same haplotype (Table 3). This pointed to the common origin of the pathogenic allele.

To confirm that the pathogenic variant is distributed in the healthy Chuvash population and other neighboring ethnic groups of the Volga-Ural region, we tested the carriage rate of the c.245A>G variant in the *GIPC3* gene by PCR-RFLP (Figure 3). In a sample of 175 healthy Chuvash individuals, four heterozygous carriers of NM_133261.3(*GIPC3*):c.245A>G (p.Asn82Ser) variant of *GIPC3* gene were revealed. The frequency of carriage of the variant in the Chuvash population was 1:44 (175/4), the population frequency was 0.0114 (95% CI 0.0031–0.0290), i.e., over 1%.

Healthy individuals from other populations neighboring the Chuvash population of the Volga-Ural region (93 Russians from the Kirov region, 320 Mari, 42 Udmurts, 183 Tatars, and 283 Bashkirs) were tested. In the tested samples, no carriers of the NM_133261.3(*GIPC3*):c.245A>G (p.Asn82Ser) variant were found. Thus, we can conclude that the c.245A>G variant in the *GIPC3* gene is exceedingly rare, if it exists, in other nearby ethnic groups and is specific to the Chuvash.

## 4. Discussion

According to the 2010 Census, ethnic Chuvash make up 67.7% (814,750 persons) of the Chuvash Republic’s population. Due to the high genetic heterogeneity of hereditary hearing pathology, the analysis of the molecular causes of hearing loss in Chuvash patients was carried out according to the developed algorithm of molecular genetic examination of NSHL in Russia and European countries.

In many populations around the world, the most common cause of NSHL is the c.35delG mutation in the *GJB2* gene, as well as some other pathogenic variants in the *GJB2*, *GJB6*, and *GJB3* genes. Therefore, at the first stage of this study, the genes of these connexins were selected as candidate genes. Both point mutations by Sanger sequencing and CNV by MLPA were analyzed. In the examined group of 26 Chuvash patients, pathogenic variants in the *GJB2*, *GJB6*, and *GJB3* genes were not detected.

Due to the high genetic heterogeneity, the next method of searching for the molecular causes of NSHL in Chuvash people was whole exome sequencing performed in a patient whose parents and siblings are also affected. A single nucleotide variant g.chr19:3586512A>G, NM_133261.3(*GIPC3*):c.245A>G (p.Asn82Ser), in the homozygous state was identified in the proband leading to the substitution of asparagine to serine p.(N82S) in the evolutionarily conservative position of the protein encoded by the *GIPC3* gene. The *GIPC3* gene encodes a protein 312 amino acid residues long that is involved in the functioning of sensitive inner ear cells and spiral ganglion neurons. Mutations in this gene are associated with nonsyndromic sensorineural hearing loss, type 15 (DFNB15) [29] with autosomal recessive inheritance. In mice, mutations of this gene are associated with nonsyndromic hearing loss or with juvenile audiogenic monogenic seizure syndrome. In patients with sensorineural hearing loss, there are biallelic nonsense, missense, and frameshifting mutations of the *GIPC3* gene [30,31,32,33,34,35]. Most of the described cases were found in Pakistan, India, and The Netherlands as single cases (Table 4). All the described patients were diagnosed with bilateral prelingual hearing loss. These mutations cause variable hearing impairment, from moderate to profound [30,31,32,33,34,35,36,37,38]. In the Russian Federation, cases of DFNB15 hearing loss have not been previously reported to date.

Although the variant is noted in the general population, its frequency is insufficient to rule out its pathogenic role. Computer prediction tools and conservativeness analysis suggest that the variant may disrupt protein function, but this information is not sufficient to determine pathogenicity. The variant reported here as the most likely cause of the disease was found by two groups of researchers [39,40] in two Iranian closely related families in three patients with NSHL in a homozygous state (Table 4).

There is some interfamily variability in the manifestations of the disease in homozygous patients of the variant (Table 1). All affected members of family #07 have profound bilateral prelingual hearing loss, starting from the first months of life. Whereas family proband #1 has severe bilateral prelingual hearing loss, the age of onset is early childhood. Proband from family #20 was diagnosed with bilateral prelingual hearing loss: complete hearing loss at right-side and severe at left-side. It should be noted that the described Iranian patients also showed interfamily and intrafamily phenotype variability. The patient described by Bitarafan F. et al. [39] had severe hearing impairment from birth; the siblings described by Kannan-Sundhari A. et al. [40] had the onset of the disease from birth, but the severity varied: one had moderate, the other, from moderate to severe (Table 4). Noncoding changes, such as variants located deep in intronic sequence regions which were out of our study, may explain the phenotype variability in carriers of NM_133261.3(*GIPC3*):c.245A>G (p.Asn82Ser) variant, although, of course, their phenotype may be explained by variants in other genes.

**Table 4 genes-12-00820-t004:** Mutations in the *GIPC3* gene associated with hearing loss described in the literature.

No.	Nucleotide Variant (Amino Acid Change)	Mutation Type	Domain	Number of Patients/Number of Families	Hearing Loss Severity	Population	Reference
1	NM_133261.3(*GIPC3*):c.122C>A (p.Thr41Lys)	Missense	GH1	1/1	Severe to profound	Saudi Arabia	[33]
2	NM_133261.3(*GIPC3*):c.136G>A (p.Gly46Arg)	Missense	GH1	1/1	Severe to profound	Pakistan	[32]
3	NM_133261.3(*GIPC3*):c.226-1G>T	Splicing site	NA1	1/1	Severe to profound	Pakistan	[35]
4	NM_133261.3(*GIPC3*):c.245A>G (p.Asn82Ser)	Missense	GH1	1/1	Severe	Iran	[39]
				2/1	Moderate; moderate to severe	Iran	[40]
5	NM_133261.3(*GIPC3*):c.264G>A (p.Met88Ile)	Missense	GH1	1/1	Mild to severe	Pakistan	[32]
6	NM_133261.3(*GIPC3*):c.281G>A (p.Gly94Asp)	Missense	GH1	1/1	Mild to severe	Pakistan	[32]
7	NM_133261.3(*GIPC3*):c.472G>A (p.Glu158Lys)	Missense	PDZ	5/1	Severe to profound	Iran (Arab origin)	[36]
8	NM_133261.3(*GIPC3*):c.508C>A (p.His170Asn)	Missense	PDZ	1/1	AR NSHL	Turkey	[34]
				1/1	AR NSHL	Turkey	[35]
9	NM_133261.3(*GIPC3*):c.565C>T (p.Arg189Cys)	Missense	PDZ	1/1	Severe to profound	Pakistan	[32]
10	NM_133261.3(*GIPC3*):c.662C>T (p.Thr221Ile)	Missense	GH2	1/1	Profound	Pakistan	[32]
11	NM_133261.3(*GIPC3*):c.685dupG (p.Ala229GlyfsTer10)	Frameshift	H2	1/1	Moderate to severe	Pakistan	[32]
12	NM_133261.3(*GIPC3*):c.767G>A (p.Gly256Asp)	Missense	GH2	1/1	Moderate to severe	Pakistan	[32]
13	NM_133261.3(*GIPC3*):c.785T>G (p.Leu262Arg)	Missense	GH2	1/1	Stable, profound	India	[31]
14	NM_133261.3(*GIPC3*):c.903G>A (p.Trp301Ter)	Nonsense	GH2	1/1	Progressive, profound	Holland	[31]
15	NM_133261.3(*GIPC3*):c.759C>G (p.Ser253Arg)	Missense	GH2	1/1	Severe	Pakistan	[37]
16	NM_133261.3(*GIPC3*):c.764T>A (p.Met255Lys)	Missense	GH2	2/1	Severe to profound	Algeria	[38]
17	NM_133261.3(*GIPC3*):c.937T>C (p.Ter313GlnextTer98)	Stop-loss	GH2	5/1	ND	Pakistan	[41]

ND—no data.

In our study, the NM_133261.3(*GIPC3*):c.245A>G (p.Asn82Ser) variant was found in 25.0% of mutant chromosomes in Chuvash patients with NSHL (13/52), in six patients in the homozygous state and in one in the heterozygous state. In the latter, no other pathogenic variants in the *GIPC3* gene were detected during Sanger sequencing, so it can be assumed that the molecular genetic causes of the disease are pathogenic variants in another gene, and heterozygous carriage is due to a high population frequency of the NM_133261.3(*GIPC3*):c.245A>G (p.Asn82Ser) variant in Chuvash population. This is supported by the fact that the heterozygous carriage was found in a healthy mother in family #13, whereas proband does not carry this variant.

Indeed, in the Chuvash population of healthy individuals, the frequency of NM_133261.3(*GIPC3*):c.245A>G (p.Asn82Ser) variant is more than 1% (0.0114, 95% CI 0.0031–0.0290; 4/350 chromosomes). While in five neighboring populations of the Volga-Ural region with a different ethnic composition (Russians of the Kirov region, Mari, Udmurts, Tatars, Bashkirs), this variant was not detected (0 out of 1842 chromosomes, frequency = 0.0000, 95% CI 0.0000–0.0016).

To explain this specifically for Chuvash population result, data on the formation of the Chuvash ethnic group are used. Based on numerous historical, cultural, and linguistic data, most researchers recognize the Chuvash as descendants of the Bulgarian and Suvar tribes that appeared on the Middle Volga in the VII–VIII centuries. The settlement of the Chuvash region by the Bulgarian-Suvars was intensive until the middle of the XIV century. The ancestral home of the local Chuvash population—the State of Volga Bulgaria, flourished until the XIII century, and its population was approaching 1.5 million. The territory of the State was inhabited by the Prachuvash, Mari (Cheremis), Udmurts, Bashkirs, Tatars, and Russians. As a result of the invasion of the Tatar–Mongol Golden Horde and the plague epidemic, at least ⅘ of the population of the state was destroyed, and it ceased to exist. Thus, the gene pool of the remaining Bulgarian population formed the basis of the modern Chuvash ethnic group. According to the published data, at the beginning of the XV century, i.e., after six generations, there were about 100 thousand Chuvash people. Official statistics on the number of Chuvash people could not be found in literary sources until the XV century. The Chuvash were known as “mountain Cheremis”. It can be assumed that by the beginning of the XIII century, after the events that took place, about 40–50 thousand Prachuvash remained in Volga Bulgaria, who mixed with other peoples of Volga Bulgaria. Until the middle of the twentieth century, 90% of the Chuvash people lived in rural areas, preserving their traditional culture and positive ethnic marital assortativeness. Over the past five centuries, the population has grown to 1.5 million people, i.e., increased by 15 times, which could possibly lead to the population effect of the “bottle neck” [42,43]. In such a situation, with social isolation and reduced migration activity of the population (which has been observed for almost three centuries), it is possible to “fix” certain alleles in the population, which caused a high accumulation of a number of hereditary diseases or specific genetic variants for the Chuvash people [19]. Among these diseases were identified: osteopetrosis, infantile malignant, type 1 (variant NM_006019.4(*TCIRG1*):c.807+5G>A, population frequency 1.68%) [44]; familial erythrocytosis, type 2 (variant NM_000551.4(*VHL*):c.598C>T (p.Arg200Trp), frequency 1.84%) [45]; hypotrichosis type 7 (variant *LIPH* EX4 DEL, frequency 2.72%) [46]; woolly hair, autosomal recessive type 3, with hypotrichosis (variant NM_181534.4(*KRT25*):c.712G>T (p.Val238Leu), frequency 1.5%) [47]; cystic fibrosis (variant NM_000492.4(*CFTR*):c.274G>A (p.Glu92Lys), frequency 0.73%) [48].

## 5. Conclusions

Thus, as a result of the conducted study, the genetic variant c.245A>G in the *GIPC3* gene was identified with high frequency. This is the molecular cause of hereditary sensorineural hearing loss in the Chuvash people, and it was previously described in patients with NSHL of Iranian origin [37,38]. Its high frequency in the Chuvash population may be due to a number of consecutive events that occurred in the historical past of the Chuvash population: the founder effect, the “bottleneck,” and/or gene drift.

## Figures and Tables

**Figure 1 genes-12-00820-f001:**
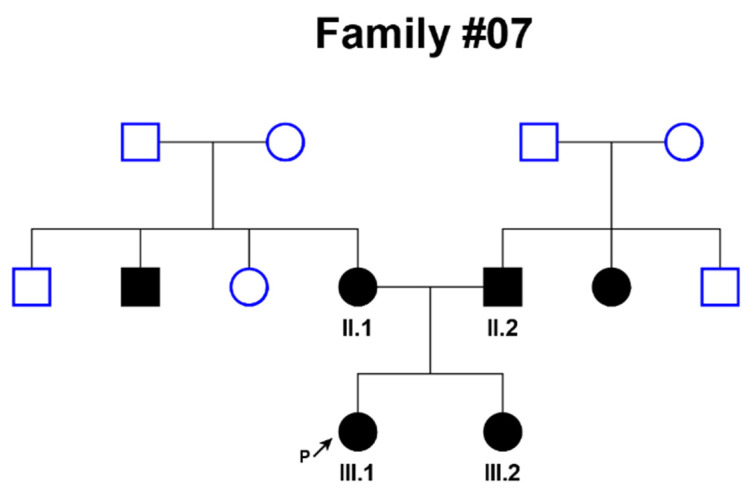
Pedigree of family #07 affected by hereditary nonsyndromic sensorineural hearing loss. P indicates proband.

**Figure 2 genes-12-00820-f002:**
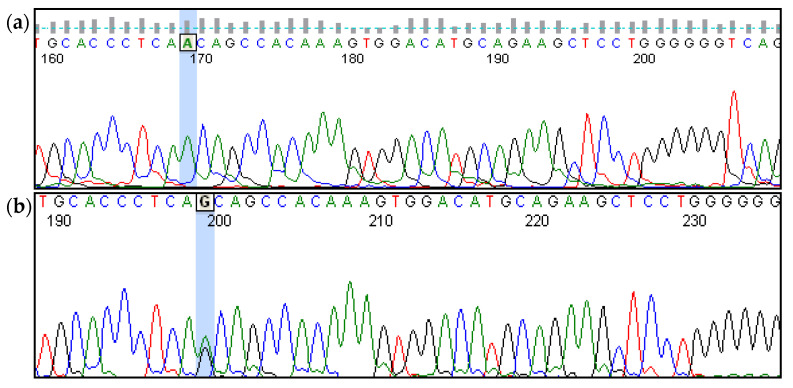
Chromatograms of sequencing results of the *GIPC3* exon 2 fragment. (**a**) Homozygous for NM_133261.3(*GIPC3*):c.245A>G (p.Asn82Ser) variant proband III.2 from family #07. (**b**) Heterozygous for NM_133261.3(*GIPC3*):c.245A>G (p.Asn82Ser) variant proband from family #04. (**c**) Homozygous for wild type proband from family #16.

**Figure 3 genes-12-00820-f003:**
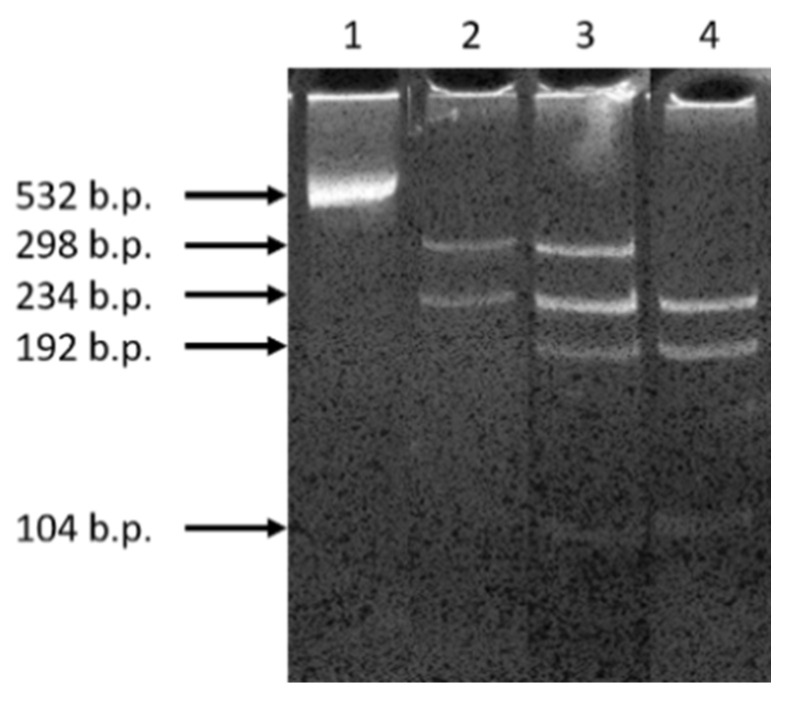
Electrophoregram of PCR-RFLP test-system performance for population screening for c.245A>G variant in *GIPC3* gene. Lanes: 1—uncut fragment; 2—wild type fragment cut with *Bst*DEI restriction endonuclease; 3—heterozygous fragment cut with *Bst*DEI; 4—homozygous fragment cut with *Bst*DEI.

**Table 1 genes-12-00820-t001:** Patients included in the study.

Family #	Patient’s Age	Age of Onset	Age of Manifestation	Degree of Hearing Loss	Genotype for *GIPC3* Coding Region
01	24	Early childhood	noticed in 2 years	Bilateral; severe	NM_133261.3(*GIPC3*):c.[245A>G];[245A>G] (p.[Asn82Ser];[Asn82Ser])
02	14	Congenital	from the first months of life	Bilateral; profound	NM_133261.3(*GIPC3*):c.[=];[=]
03	8	Congenital	from the first months of life	Bilateral; severe.	NM_133261.3(*GIPC3*):c.[=];[=]
04	14	Congenital	at 5 months	Bilateral; complete	NM_133261.3(*GIPC3*):c.[245A>G];[=] (p.[Asn82Ser];[=])
05	18	Congenital	noticed in 2 years	Bilateral; profound	NM_133261.3(*GIPC3*):c.[=];[=]
06	33	Congenital	noticed in 2 years	Bilateral; severe	NM_133261.3(*GIPC3*):c.[=];[=]
07	1	Congenital	from the first months of life	Bilateral; profound	NM_133261.3(*GIPC3*):c.[245A>G];[245A>G] (p.[Asn82Ser];[Asn82Ser])
07	28	Congenital	from the first months of life	Bilateral; profound	NM_133261.3(*GIPC3*):c.[245A>G];[245A>G] (p.[Asn82Ser];[Asn82Ser])
07	32	Congenital	from the first months of life	Bilateral; profound	NM_133261.3(*GIPC3*):c.[245A>G];[245A>G] (p.[Asn82Ser];[Asn82Ser])
07	4	Congenital	from the first months of life	Bilateral; profound	NM_133261.3(*GIPC3*):c.[245A>G];[245A>G] (p.[Asn82Ser];[Asn82Ser])
08	12	Congenital	noticed in 3 years	OD-moderate, OS severe	NM_133261.3(*GIPC3*):c.[=];[=]
09	10	Congenital	noticed in 1 year	OD-moderate, OS severe	NM_133261.3(*GIPC3*):c.[=];[=]
10	18	Congenital	from the first months of life	Bilateral; profound	NM_133261.3(*GIPC3*):c.[=];[=]
11	14	Congenital	from the first months of life	OD-severe, OS profound	NM_133261.3(*GIPC3*):c.[=];[=]
12	6	Congenital	noticed in 3 years	Bilateral; profound.	NM_133261.3(*GIPC3*):c.[=];[=]
13	10	Congenital	from the first months of life	OD-severe, OS profound	NM_133261.3(*GIPC3*):c.[=];[=]
14	15	Congenital	from the first months of life	Bilateral; profound	NM_133261.3(*GIPC3*):c.[=];[=]
15	16	Congenital	noticed in 1.5 years	Bilateral; profound	NM_133261.3(*GIPC3*):c.[=];[=]
16	19	Congenital	from the first months of life	Bilateral; complete	NM_133261.3(*GIPC3*):c.[=];[=]
17	14	Childhood	5–6 years	Unilateral; OD moderate	NM_133261.3(*GIPC3*):c.[=];[=]
17	42	Childhood	13–14 years	Unilateral; OS moderate to severe	NM_133261.3(*GIPC3*):c.[=];[=]
18	13	Congenital	from the first months of life	Bilateral; profound	NM_133261.3(*GIPC3*):c.[=];[=]
18	38	Congenital	noticed in 1.5 years	Bilateral; profound	NM_133261.3(*GIPC3*):c.[=];[=]
19	36	Congenital	noticed by 6–7 months	OD complete, OS profound	NM_133261.3(*GIPC3*):c.[=];[=]
20	22	Congenital	from the first months of life	Bilateral; profound	NM_133261.3(*GIPC3*):c.[245A>G];[245A>G] (p.[Asn82Ser];[Asn82Ser])
21	22	Congenital	noticed by 12–13 months	OD moderate, OS severe	NM_133261.3(*GIPC3*):c.[=];[=]

**Table 2 genes-12-00820-t002:** Primers used in the study.

Gene	Primer Name and Sequence in 5′—3′ Direction
*GJB2*	F1 TCATGGGGGCTCAAAGGAAC
	R1 AAGGACGTGTGTTGGTCCAG
	F2 GTTCTGTCCTAGCTAGTGATT
	R2 GGTTGCCTCATCCCTCTCAT
*GJB3*	F1 CGTTGTGAGTATTGAACAAGTCAGAACTCAG
	R1 GTTGATCCCTTCCTGGTTA
	F2 CTCTGCTACCTCATCTGCCA
	R2 GTTGATCCCTTCCTGGTTGA
*GJB6*	F1 CTTTCAGGGTGGGCATTCCT
	R1 AGCACAACTCTGCCACGTTA
	F2 CTTCGTCTGCAACACACTGC
	R2 GCAATGCTCCTTTGTCAAGCA
*GIPC3*-ex1	F1 CTTATTTGTGGTCCCTGTTCTTC
	R1 AGTCCTAAGACCTGCCCATCT
*GIPC3*-ex2–4	F2 CTCTCTCTGTTCTGGGGGTCC
	R2 ACCTACGAGTTTCTGATACCCTG
*GIPC3*-ex5–6	F3 GGCATGGAACTGGGATGTTA
	R3 GCACATAGCTTGGCCTCAGAT
*GIPC3*-c.245A>G	GIPC3-Fmut TCTCCACCTGCTGGAAGTCT
	GIPC3-Rmut CCTCGATCCGGTTGATGAT

**Table 3 genes-12-00820-t003:** Haplotype linked to the c.245A>G variant in *GIPC3* gene.

Single Nucleotide Polymorphism (SNP)	rs112835547	rs34722692	rs8100350	rs8113232	rs4806942	rs10406702	rs10426399	rs28532669	rs78077103	rs78077103	rs17348907
Possible genotypes (ref/alt)	C/T	C/T	G/A	A/G	A/G	T/C	C/T	G/A	G/T	G/A	C/T
Haplotype linked to the variant	T	C	G	G	G	T	C	G	G	G	T

## Data Availability

The datasets used and/or analyzed during the current study are available from the corresponding author on reasonable request.

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
