# Peer review of "Genetic Variant c.245A>G (p.Asn82Ser) in GIPC3 Gene Is a Frequent Cause of Hereditary Nonsyndromic Sensorineural Hearing Loss in Chuvash Population"

_genes, 2021, doi:10.3390/genes12060820_

Round 1

Reviewer 1 Report

Dear authors,

compliment to your work. i think genetical testing regarding hearing in small populations is important.

I do have some questions, advices to improve your papper:

  • I would like to know the number of population of Chuvash population, (there are no lines in the article)
  • hearing loss should be in dB, measured by pure tone audiogram and erxpresed like mild, severe, profound.... as in the citated literature (not 3-4 degrees)
  • did you excluded inner ear annomalies in the cases where no genetic background was found?

Author Response

First of all, we eager to acknowledge the Reviewer for deep analysis of the manuscript. Please see below our responses to the Reviewer's comments.

  1. I would like to know the number of population of Chuvash population, (there are no lines in the article)

According to the 2010 Census, ethnic Chuvash make up 67.7% (814,750 persons) of the Chuvash Republic's population. (added to the beginning of Discussion) 

  1. hearing loss should be in dB, measured by pure tone audiogram and erxpresed like mild, severe, profound.... as in the citated literature (not 3-4 degrees) – Thanks, we have changed the division of the disease severity, adopted in the Russian Federation, for division according to WHO and corrected Table 1 and abstract before Table 4.

the World Health Organization (WHO) grades hearing impairment as mild (instead of 1 degree) (26–40 dB HL), moderate (41–60 dB HL) (2 degree), severe (61–80 dB HL) (3 degree) or profound (81 dB HL or greater) (4 degree) in the better ear.

  • . did you excluded inner ear annomalies in the cases where no genetic background was found?
  • Yes, when examined by specialists, anomalies of the inner ear were excluded in all patients, including those who did not have pathogenic variants, but we do not discuss this issue further in this article, since it is devoted to the variant in the GIPC3 gene.

Reviewer 2 Report

The manuscript by Petrova et al. entitled “Genetic Variant c.245A>G (p.Asn82Ser) in GIPC3 Gene is a Frequent Cause of Hereditary Non-syndromic Sensorineural Hearing Loss in Chuvash Population” reports the results of screening of 21 unrelated families. The authors detected a single variation c.245A>G (p.Asn82Ser) in GIPC3 in a trio of family #07 through NGS and then sequenced this variant in affected individuals of other 20 families. They found the same variant in homozygous form in two families #01 &20) while in one family (#04) in heterozygous form associated with hearing loss. Further, the authors screened the same variant in 175 ethnically matched healthy individuals and found in four individuals in heterozygous carrier form. They have described the founder effect of this variant in Chuvash Population in Russia. Data being reported here will make a useful addition to already published information.

However, minor changes are recommended to further improve the quality of this manuscript.

  1. In materials and methods section on line 83 it is written “21 Chuvash families ascertained” while on line 88 it says “26 patients with NSHL from 22 unrelated families were examined”. This discrepancy should be removed.
  2. On line 154 it is says “Most pathogenicity predictions tools”, should mention some tools used.
  3. Nomenclature of the variation in the whole manuscript is incomplete and authors should provide complete variant nomenclature.
  4. At the end of document, it is written the date of Institutional Review Board “protocol No. 17/2006 dated 02.02.2006”. Are they still using a 15 year old protocol and did not revise it since then?
  5. The references used in the manuscript to describe prevalence (ref #1-3) are more than ten year old. They can use instead e.g., PMID: 30047343. There are other references which should be included.

Author Response

First of all we eager to acknowledge the Reviewer for the detailed analysis of our manuscript. Please find our responses to the Reviewer's comments.

  1.    In materials and methods section on line 83 it is written “21 Chuvash families ascertained” while on line 88 it says “26 patients with NSHL from 22 unrelated families were examined”. This discrepancy should be removed. –       Thanks, we have corrected this mistake.
  2.  On line 154 it is says “Most pathogenicity predictions tools”, should mention some tools used. We have changed to “Most pathogenicity predictions tools, such as SIFT (https://sift.bii.a-star.edu.sg/), Polyphen2 (http://genet ics.bwh.harva rd.edu/pph2/), and MutationTaster (http://www.mutationtaster.org/), classify the variant as damaging.”
  3. Nomenclature of the variation in the whole manuscript is incomplete and authors should provide complete variant nomenclature. - We have applied a single nomenclature for variants throughout the text and tables.
    NM_133261.3(GIPC3):c.245G>A(p.Asn82Ser) and Tables 1, 4
    NM_006019.4(TCIRG1):c.807+5G>A
    NM_000551.4(VHL):c.598C>T (p.Arg200Trp)
    LIPH, EX4 DEL
    NM_181534.4(KRT25):c.712G>T (p.Val238Leu)
    NM_000492.4(CFTR):c.274G>A (p.Glu92Lys) 
  4. At the end of document, it is written the date of Institutional Review Board “protocol No. 17/2006 dated 02.02.2006”. Are they still using a 15 year old protocol and did not revise it since then? – We used both old protocol and revised protocol protocol No. 5/2010 dated 20.12.2010.
  5. The references used in the manuscript to describe prevalence (ref #1-3) are more than ten year old. They can use instead e.g., PMID: 30047343. There are other references which should be included. – We have changed references to:
    1.Korver AM, Smith RJ, Van Camp G, Schleiss MR, Bitner-Glindzicz MA, Lustig LR, Usami SI, Boudewyns AN. Congenital hearing loss. Nat Rev Dis Primers. 2017;12(3):16094. https://doi: 10.1038/nrdp.2016.94.
    2.Markova TG, Bliznetz EA, Polyakov AV, Tavartkiladze GA. Twenty years of clinical studies of GJB2-linked hearing loss in Russia. Vestn Otorinolaringol. 2018;83(4):31-36. https://doi:10.17116/otorino201883431. In Russ. 
    3.Meena R, Ayub M. Genetics of human hereditary hearing impairment. J Ayub Med Coll Abbottabad. 2017;29(4):671-676.
    5.Yang T, Guo L, Wang L, Yu X. Diagnosis, Intervention, and Prevention of Genetic Hearing Loss. Adv Exp Med Biol. 2019;1130:73-92. https://doi: 10.1007/978-981-13-6123-4_5.